# Confidence reports in decision-making with multiple alternatives violate the Bayesian confidence hypothesis

Hsin-Hung Li [1✉] & Wei Ji Ma [1,2]

Decision confidence reflects our ability to evaluate the quality of decisions and guides subsequent behavior. Experiments on confidence reports have almost exclusively focused on two-alternative decision-making. In this realm, the leading theory is that confidence reflects the probability that a decision is correct (the posterior probability of the chosen option). There is, however, another possibility, namely that people are less confident if the best two options are closer to each other in posterior probability, regardless of how probable they are in absolute terms. This possibility has not previously been considered because in two-alternative decisions, it reduces to the leading theory. Here, we test this alternative theory in a three-alternative visual categorization task. We found that confidence reports are best explained by the difference between the posterior probabilities of the best and the next-best options, rather than by the posterior probability of the chosen (best) option alone, or by the overall uncertainty (entropy) of the posterior distribution. Our results upend the leading notion of decision confidence and instead suggest that confidence reflects the observer's subjective probability that they made the best possible decision.

[1] Department of Psychology, New York University, New York, NY, USA. [2] Center for Neural Science, New York University, New York, NY, USA.
✉email: hsin.hung.li@nyu.edu

**C**onfidence refers to the "sense of knowing" that comes with a decision. Confidence affects the planning of subsequent actions after a decision[1,2], learning[3], and cooperation in group decision making[4]. Failures of utilizing confidence information have been linked to psychiatric disorders[5].

While human observers can report their self-assessment of the quality of their decisions[6–12], the computations underlying confidence reports are still insufficiently understood. The leading theory of confidence suggested that confidence reflects the probability that a decision is correct[7,8,13–17]. We refer to this idea as the "Bayesian confidence hypothesis", meaning that the decision-makers use the posterior probability of the chosen category (i.e. the subjective probability that decision is correct) for their confidence reports. Accordingly, in neurophysiological studies, a brain region or a neural process is considered to represent confidence if its responses correlate with the probability that a decision is correct[18–20]. Behavioral studies testing whether human confidence reports follow Bayesian confidence hypothesis have shown mixed results: While some studies found resemblances between Bayesian confidence and empirical data[18,19,21,22], others have suggested that confidence reports deviate from the Bayesian confidence hypothesis[23–25].

Even though the Bayesian confidence hypothesis is the leading theory of confidence, there is currently no evidence to rule out the possibility that confidence is affected by the probability of correct of the unchosen options. Specifically, people could be less confident if the next-best option is very close to the best option. In other words, confidence could depend on the *difference* between the posterior probabilities of the best and the next-best options, rather than on the absolute value of the posterior of the best option. The reason that this idea has not been tested before might be that previous studies of decision confidence predominantly used two-alternative decision tasks; in such tasks, the alternative hypothesis is equivalent to the Bayesian confidence hypothesis, because the difference between the two posterior probabilities in a two-alternative task is a monotonic function of the highest posterior probability. Thus, to dissociate these two models of confidence, we need more than two alternatives. Here, we use a three-alternative decision task. To preview our main result, we find that

the difference-based model accounts well for the data, whereas the model corresponding to the Bayesian confidence hypothesis and a third, entropy-based model do not.

To investigate the computations underlying confidence reports in the presence of multiple alternatives, we designed a three-alternative categorization task. On each trial, participants viewed a large number of exemplar dots from each of the three categories (color-coded), along with one target dot in a different color (Fig. 1a). Each category corresponded to an uncorrelated, isotropic Gaussian distribution in the plane. We asked participants to regard the stimulus as a bird's eye view of three groups of people. People within a group wear shirts of the same color, and the target dot represents a person from one of the three groups. Participants made two responses: the category of the target, and their confidence in their decision on a four-point Likert scale.

To manipulate participants' beliefs (posterior probability distribution), we used different configurations of the category distributions and varied the position of the target dot within each configuration (Fig. 1b, c). This design allowed us to test quantitative models of how the posterior distribution gives rise to confidence reports (see an illustration of this idea in Supplementary Fig. 1).

## Results

**Model**. Generative model. Each category is equally probable. We assume that the observer makes a noisy measurement $x$ of the position $s$ of the target dot. We model the noise as obeying an isotropic Gaussian distribution centered at the target dot.

Decision model: We now consider a Bayesian observer. We assume that the observer knows that each category is equally probable ($p(C) = 1/3$), and knows the distribution associated with each category (group) based on the exemplar dots. Given a measurement $x$, the posterior probability of category $C$ is then

$$p(C|\mathbf{x}) = \frac{p(\mathbf{x}|C)}{\sum_{C'=1}^{3} p(\mathbf{x}|C')} \qquad (1)$$

We further assume that due to decision noise or inference noise, the observer does not maintain the exact posterior

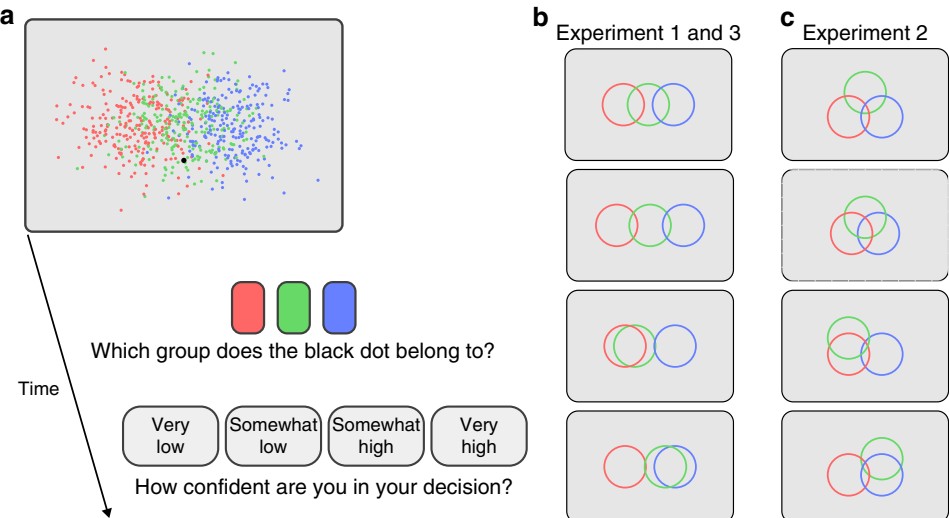

**Fig. 1 Experimental procedure and stimuli. a** Each trial started with the presentation of the stimulus including exemplar dots in three different colors representing the distribution of each of the three categories and one target dot, the black dot. Observers first reported their decisions in the categorization task and then reported their confidence by using the rectangular buttons presented at the bottom of the screen. **b, c** Schematic representation of the distribution of the categories. The circles are centered at the mean location of each category. The width of the circles corresponds to 2.5 times the standard deviation of the category distribution. (**b**) The four conditions tested in Experiment 1 and 3. **c** The four conditions tested in Experiment 2. The exemplar dots in (**a**) are based on the distribution depicted in the top panel in (**b**).

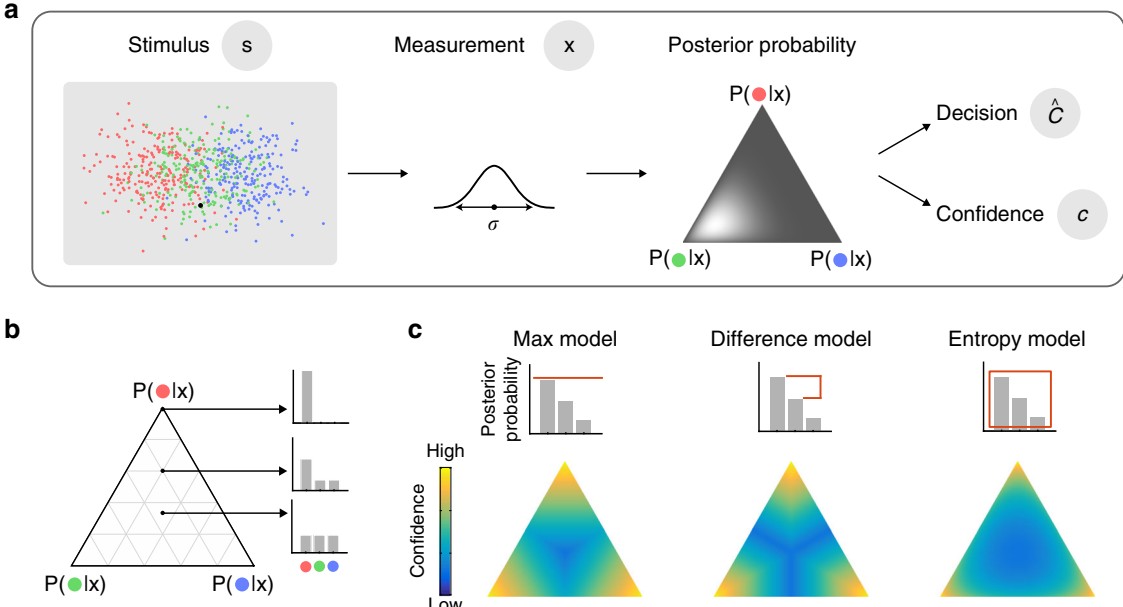

**Fig. 2 Models. a** Generative model. Target position is represented by **s**. Two sources of variability are considered in the model: First, observers have access to noisy measurement **x**, a Gaussian distribution centered at **s** with a standard deviation $\sigma$. Second, given the same measurement **x**, the posterior distribution varies across trials due to decision noise, modeled by Dirichlet distribution, of which spread (represented by the shade of the ternary plot) is controlled by a parameter $\alpha$ (see Methods). On each trial, a decision $\hat{C}$ and a confidence c are read out from the posterior distribution of that trial. **b** We use ternary plots to represent all possible posterior distributions. For example, a point at the center represents a uniform posterior distribution; at the corners of the ternary plot, the posterior probability of one category is one while the posterior for the other two categories are zeros. **c** The bar graphs illustrate how confidence is read out from posterior probabilities in each model. For the purposes of these plots, we did not include decision noise here. The color of each ternary plot represents the confidence as a function of posterior distribution for each model. The color is scaled for each ternary plot (independently) to take the whole range of the color bar.

distribution, $p(C \mid \mathbf{x})$, but instead a noisy version of it. This type of decision noise is consistent with the notion that a portion of variability in behavior is due to "late noise" at the level of decision variable[26–28]. We modeled decision noise by drawing a noisy posterior distribution from a Dirichlet distribution around the true posterior (Fig. 2a, b; See details in Methods). In our case, the true posterior, which we denote by **p**, consists of the three posterior probabilities from Eq.(1): $\mathbf{p} = (p(C = 1 \mid \mathbf{x}), p(C = 2 \mid \mathbf{x}), p(C = 3 \mid \mathbf{x}))$. The magnitude of decision noise, the amount of variation around **p**, is (inversely) controlled by a concentration parameter $\alpha > 0$. When $\alpha \to \infty$, the variation vanishes and the posterior is noiseless. In general, the "noisy posterior", which we denote as a vector **q**, satisfies $\mathbf{q} \sim \text{Dirichlet}(\alpha \mathbf{p})$. We assume that when reporting the category of the target, the observer chooses the category $C$ with the highest $q(C \mid \mathbf{x})$. Unless otherwise specified, we will from now on refer to the noisy posterior distribution as simply the posterior distribution.

We introduce three models of confidence reports: the Max model, the Entropy model and the Difference model. Each of these models contains two steps: (a) mapping the posterior distribution (**q**) to a real-valued internal confidence variable; (b) applying three criteria to this confidence variable to divide its space into four regions, which, when ordered, map to the four confidence ratings. The second step accounts for every possible monotonic mapping from the internal confidence variable to the four-point confidence rating. The three models differ only in the first step.

The Max model corresponds to the Bayesian confidence hypothesis. In this model, the confidence variable is the probability that the chosen category is correct, or in other words, it is the highest of the three posterior probabilities (Fig. 2c). In this model, the observer is least confident when the posterior distribution is uniform. Importantly, after the posterior distribution is computed,

the posterior probability of the unchosen options does not further contribute to the computation of confidence.

In the Difference model, the confidence variable is the difference between the highest and second-highest posterior probabilities. In this model, confidence is low if the evidence for the next-best option is strong, and the observer is least confident whenever the two most probable categories are equally probable. One interpretation of this model is that confidence reflects the observer's subjective probability that they made the best possible choice, regardless of the actual posterior probability of that choice. An alternative interpretation is that decision-making consists of an iterative process in which the observer reduces a multiple-alternative task to simpler (two-alternative) tasks (see the Discussion section). (Note that a model that uses the difference of the probability of the best option and the average of the non-chosen options is equivalent to the Max model.)

In the Entropy model, the confidence variable is the negative of the uncertainty conveyed by the entire posterior distribution, as quantified by its negative entropy. High confidence is associated with low entropy, and vice versa. Like in the Max model, the observer is least confident when the posterior distribution is uniform. Unlike in the Max model, however, the posterior probabilities of the non-chosen categories directly affect confidence. For the details of the models, see Methods.

All three models are Bayesian in the sense that they compute the posterior probability distribution and categorize the target dot into the category with the highest posterior. Thus, in all three models, the unchosen options "implicitly" affect confidence by contributing to the denominator in the computation of the posterior probabilities. In the Discussion, we discuss a model in which the unchosen option (e.g., the least probable category) is disregarded before even contributing to the normalization of the posterior. The three models differ in how the confidence variable

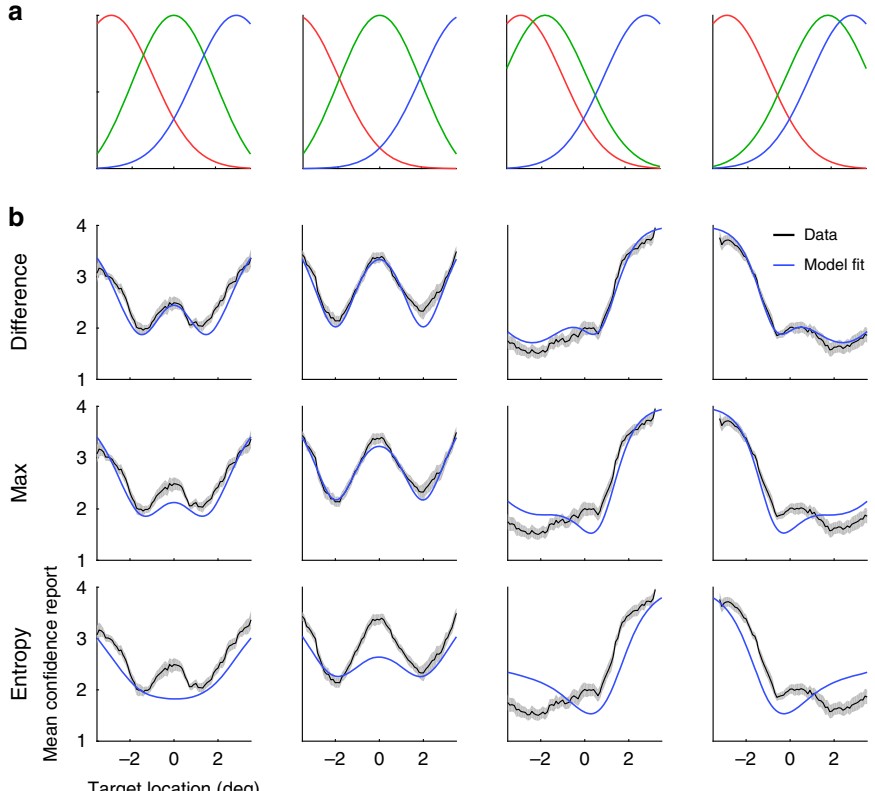

**Fig. 3 Experiment 1. a** The distribution of the reference dots in each condition. **b** Mean confidence report as a function of target position for each of the four conditions. The black curves represent group mean ± 1s.e.m. Blue curves represent the model fit averaged across individuals.

is read out from the posterior distribution. The Max model is unique in assuming that after the computation of the posterior probabilities, the unchosen categories do not further affect the computation of confidence.

In our three-alternative task, these models generate qualitatively different mappings from the posterior distribution to the confidence variable (Fig. 2c). In a standard two-alternative task, however, the models would have been indistinguishable, because the probability of the chosen category would have determined the probability of the non-chosen category.

The Max, Difference and Entropy models are our three main models. So far, the sources of variability in these models are sensory noise (Sen) and Dirichlet decision noise (Dir). We name the corresponding models Max-Sen-Dir, Diff-Sen-Dir, Ent-Sen-Dir models in the supplementary figures and supplementary tables in order to distinguish them from model variants that consider different sources of variability (introduced later).

We fitted the free parameters to the data of each individual subject using maximum-likelihood estimation, where the data on a given trial consist of a decision-confidence pair. Thus, we accounted for the joint distribution of decisions and confidence ratings[24,25,29] (see Methods). We compared models using the Akaike Information Criterion (AIC[30]). A model recovery analysis suggests that if the true model is among our tested models, our model comparison procedure is able to identify the correct model (see Methods and Supplementary Fig. 3).

**Experiment 1**. In Experiment 1, the centers of the three category distributions were aligned vertically (Fig. 1b). There were four conditions: In the first two conditions, the centers were evenly spaced horizontally. In the last two conditions, the center of the central distribution was closer to the center of either the left or the right distribution. The vertical position of the target dot was

sampled from a normal distribution, and the horizontal position of the target dot was sampled uniformly between the center of the leftmost and right-most classes plus an extension to the left and the right (see Methods).

We plotted the psychometric curves (mean confidence report as a function of the horizontal position of the target dot) by averaging confidence reports across trials using a sliding window (Fig. 3). Mean confidence report varied as a function of the horizontal position of the target. In the first two conditions (Fig. 3), where the three distributions were evenly spaced, the psychometric curves showed two dips, with the lowest confidence attained at two positions symmetric around 0°.

We simulated the predicted psychometric curves using the best-fitting parameters of each model (Fig. 3b). The fits of the Max and the Difference models resembled the data, but the best fit of the Entropy model showed a dip at the center in the first condition.

In the third and fourth conditions, in which the three distributions were unevenly spaced, mean confidence was lowest around the centers of the two distributions that were closest to each other. Only the Difference model exhibited this pattern, while the Max and the Entropy models deviated more clearly from the data.

The models not only make predictions for confidence reports, but also for the category decisions (Supplementary Fig. 2). Participants categorized the target dot based on its location, and when the target dot was close to the boundary between two neighboring categories (the location where two categories have equal likelihood), they assigned the target to those two neighboring categories with nearly equal probabilities. In general, this pattern is consistent with an observer who chooses the category associated with the highest posterior probability. The Entropy model fits worst, even though all three models used the

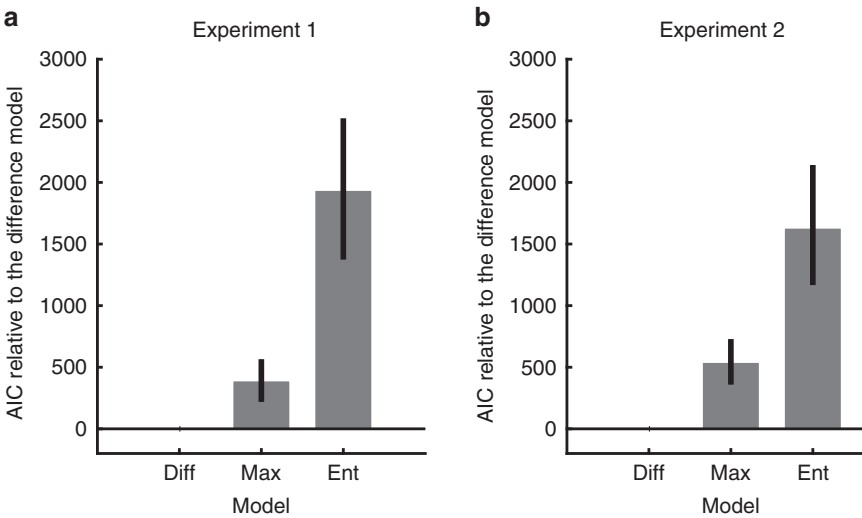

**Fig. 4 Model comparisons using ΔAIC: AIC of each model compared with the Difference model.** The bars represent ΔAIC summed across participants. The error bars represent 95% bootstrapped confidence interval. **a** Experiment 1. **b** Experiment 2.

same rule for the category decision; this is because the confidence data also need to be accounted for. The Difference model outperformed the Max model by a group-summed AIC score of 391 (95% CI [222, 569]) and the Entropy model by 1937 (95% CI [1363, 2562]) (Fig. 4a and Supplementary Table 1).

We further tested reduced versions of each of the three confidence models by removing either the sensory noise or the decision noise from the model. The Difference model outperformed the Max model and the Entropy model regardless of these manipulations (Supplementary Fig. 4A and Supplementary Table 1). The sensory noise played a minor role in this task compared to the decision noise. For example, removing the sensory noise from the Difference model increased the AIC by 121 (95% CI [48, 199]), while removing the inference noise increased the AIC by 737 (95% CI [590, 914]). Using the Bayesian information criterion (BIC)[31] for model comparison led to the same conclusions (Supplementary Fig. 5A and Supplementary Table 2).

So far, we jointly fitted the category decision and confidence reports. One could wonder whether independently fitting the confidence reports would lead to different results. We found the same results when only fitting the confidence reports: The Difference model outperformed the other two models, and the decision noise had a stronger influence on the model fit (Supplementary Figs. 6 and 7). Because the Max, the Difference and the Entropy used the same rule for category decisions, we compared category decision models that used the same decision rule (reporting the category with the highest posterior probability), but included sensory noise only, decision noise only, or both. We fitted the category decisions alone and found that the models including the decision noise fit the data better than the model with the sensory noise alone (Supplementary Figs. 8 and 9). This is similar to the results obtained by fitting the confidence reports alone or by jointly fitting both category decisions and confidence reports.

We tested various alternative models (see details in Supplementary Information). We found that the Difference model outperformed the Max and the Entropy models when we replaced Dirichlet decision noise by drawing samples from the true posterior, or when we added noise in the measurement of the category means (Supplementary Fig. 10A). In addition, we tested heuristic models that made category decisions and confidence reports based on the category means and the noisy measurement

of the target location (**x**) but did not compute posterior probabilities. Still, the heuristic models did not fit the data better than the Difference model (Supplementary Fig. 10A).

**Experiment 2.** In Experiment 2, we aimed to test whether the findings in Experiment 1 could be generalized to other stimulus configurations, where the centers of the categories varied in a two-dimensional space. We tested four conditions in which the centers of the three groups varied along both horizontal and vertical axis (Fig. 1c). We sampled the target dot positions uniformly within a circular area centered on the screen. In addition, the distribution of the categories used in Experiment 2 allowed us to probe confidence reports in a wider range of posterior distributions (Supplementary Fig. 1B). For example, we can probe the confidence report when the target dot had the same distance to all three categories in Experiment 2, but not in Experiment 1.

The "psychometric curve" is now a heat map in two dimensions (Fig. 5). The fits to these psychometric curves showed different patterns among the three models: When the three groups formed an equilateral triangle (Fig. 5, the first and second columns), the confidence (as a function of target location) estimated by the Entropy model exhibited contours that were more convex than that in the data. In the last two conditions (Fig. 5, the third and fourth columns), compared to the other two models, the Difference model showed stronger resemblance to the data, as the model exhibited an extended low confidence region at the side where two categories were positioned closely. The results of model comparisons were consistent with Experiment 1. The Difference model outperformed the Max model by a group-summed AIC score of 541 (95% CI [371, 735]) and the Entropy model by 1631 (95% CI [1179, 2159]) (Fig. 4b). The model with both sensory and inference noise explained the data the best, and the inference noise had a stronger influence on the model fit than the sensory noise (Supplementary Fig. 4B, Supplementary Fig. 5B, Supplementary Tables 1 and 2).

Consistent with Experiment 1, we found that the Difference model outperformed the Max and the Entropy model when we only fitted the confidence reports (Supplementary Fig. 6B). Models that considered other sources of variability or used heuristic decision rules did not perform better than the Difference model (Supplementary Fig. 10B).

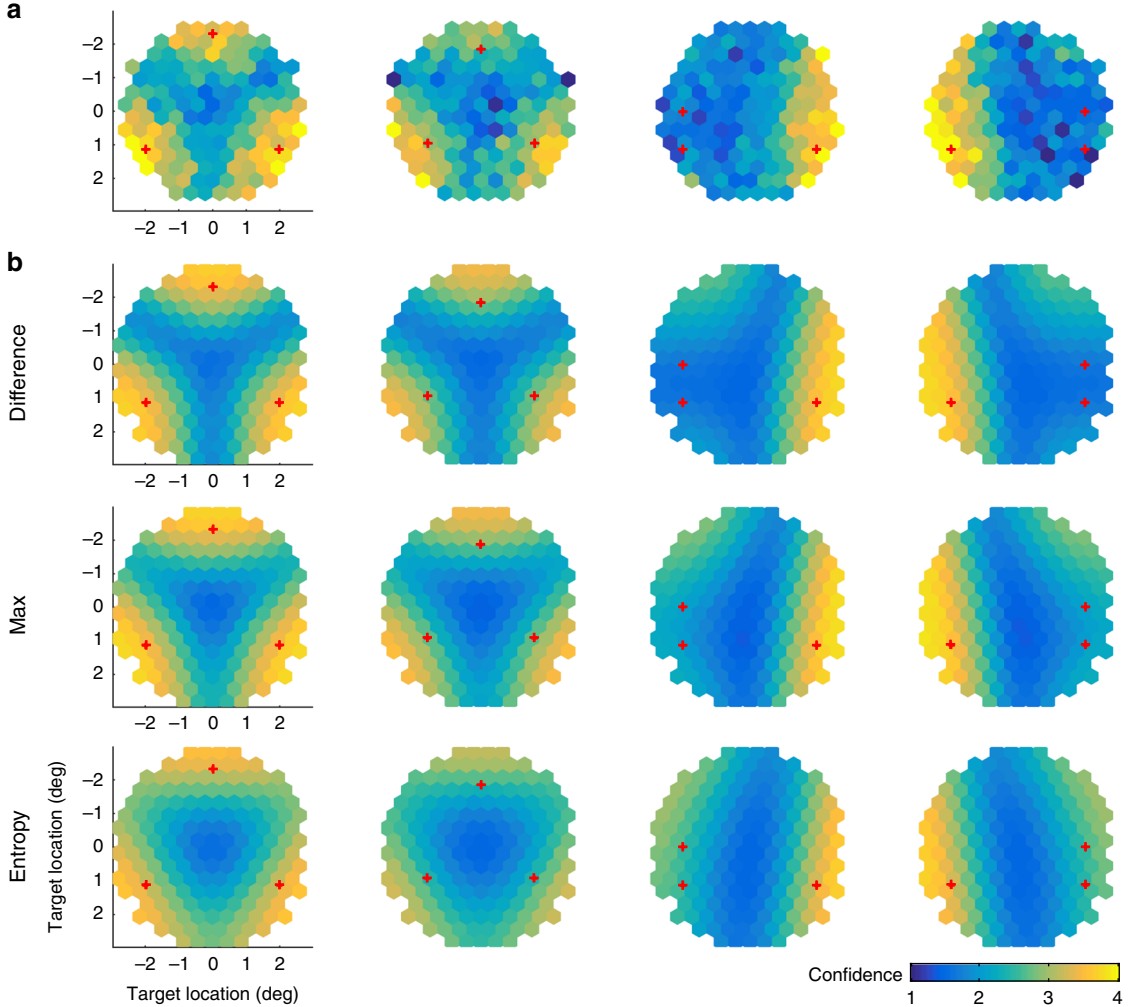

**Fig. 5 Experiment 2. a** The mean confidence report as a function of target positions. **b** Model fit averaged across individuals. The red crosses in each panel represent the center of each of the three categories.

**Experiment 3**. So far, we found that the Difference model fits the data better than the Max model and the Entropy model. However, whether participants report the probability that a decision is correct (the Max model) might depend on the experimental design. In Experiment 1 and 2, participants received no feedback on their category decision. Thus, the probability of being correct in the task could be difficult to learn. To investigate this issue, in Experiment 3, using the same four stimulus configurations as those in Experiment 1 (Fig. 1b), we randomly chose one of the three groups as the true target category in each trial, and sampled the target position from the distribution of the true category. Feedback was presented at the end of each trial, informing participants of the true category.

The results of model comparison were consistent with Experiment 1 and 2. The Difference model outperformed the Max model by a group-summed AIC score of 100 (95% CI [46, 156]) and the Entropy model by 1113 (95% CI [817, 1447]) (Supplementary Fig. 11, Supplementary Tables 1 and 2). The model with both sensory and inference noise explained the data the best, and the inference noise had a stronger influence on the model fit than the sensory noise (Supplementary Figs. 4C and 5C). These results held when we fitted the confidence reports alone, or when other sources of variability were considered (Supplementary Figs. 6C and 10C). Heuristic models did not fit the data better than the Difference model (Supplementary Fig. 10C).

## Discussion

To distinguish the leading model of perceptual confidence (the Bayesian confidence hypothesis) from a new alternative model in which confidence is affected by the posterior probabilities of unchosen options, we studied human confidence reports in a three-alternative perceptual decision task. We found that confidence is best described by the Difference model, in which confidence reflects the difference between the strength of observers' belief (posterior probability) of the top two options in a decision. The Max model (which corresponds to the Bayesian confidence hypothesis) and the Entropy model (in which confidence is derived from the entropy of the posterior distribution) fell short in accounting for the data. Our results were robust under changes of stimulus configurations (Experiment 1 and 2), and when trial-by-trial feedback was provided (Experiment 3). Our results demonstrate that the posterior probabilities of the unchosen categories impact confidence in decision-making.

Decision tasks with multiple alternatives not only allow us to dissociate different computational models of confidence, they are also ecologically important. In the real world, human and other animals often face decisions with multiple alternatives, such as identifying the color of a traffic light, recognizing a person, categorizing a species of an animal, or making a medical diagnosis.

Our models can be generalized to categorical choice with more than three alternatives. Specifically, the Difference model predicts that besides the posterior probabilities of the top two options, the posterior of the other options does not matter as long as they add up to the same total. A special type of categorical choice is when the world state variable is continuous (e.g., in an orientation estimation task) but gets discretized for the purpose of the experiment. Consider the specific case that the posterior distribution is Gaussian. An observer following the Difference model would compute the difference between the posteriors of the two discrete options closest to the peak. This serves as a coarse approximation to the curvature of the posterior distribution at its peak, which, for Gaussians, is monotonically related to its inverse variance, consistent with an earlier model by van den Berg et al.[29], in which confidence is based on the precision parameter of the posterior in continuous estimation tasks[29]. Outside the realm of Gaussian and similar distributions, the Difference model and van den Berg et al.[29]'s model might be distinguishable. For example, when the posterior distribution is bimodal, with the modes slightly different in height, the variance of the posterior is dominated by the separation between the modes, whereas the Difference model will use the difference in height for confidence reports.

Although many behavioral studies have emphasized similarities between human confidence reports and predictions of Bayesian models e.g[18,21,22]., the Bayesian confidence hypothesis has been questioned before[8,13–16]. In addition to the probability of being correct, confidence is influenced by various factors such as reaction time[32], post-decision processing[33–36], and the magnitude of positive evidence[37–40]. Two model comparison studies have shown deviations from Bayesian confidence hypothesis in two-alternative decision tasks[24,25]. However, in one study[24], the experimental design did not allow the authors to strongly distinguish the model that was based on Bayesian confidence hypothesis from those that were not. Moreover, in both studies[24,25], the alternative models were based on heuristic decision rules without a broader theoretical interpretation. Here, we have identified a type of deviation from the Bayesian confidence hypothesis that is not only of a qualitatively different nature, but that also raises new theoretical questions.

Specifically, the Difference model is currently a descriptive model. We have two suggestions to interpret it as an outcome of approximate inference. First, the Difference model might be an approximation to a model in which confidence depends on the probability that an observer made the best possible decision. In this view, the observer possesses metacognitive knowledge that their decision is based on the noisy posterior **q** rather than the true posterior **p**, and consequently, realizes that it is possible that the chosen category is not the category with the highest true posterior probability. Confidence would then be derived from the probability that the chosen category has the highest true posterior probability. The stronger the evidence for the next-best option, the less likely is the case, which would lead to lower confidence. This interpretation is consistent with recent work that showed that subjective confidence guides information seeking during decision-making[41]. Under the Difference model, during information seeking, the observer's goal is to make sure that the best option is better than the alternative options. Low confidence would encourage the observer to collect more information in order to strengthen the belief that the best option is better than the next-best option.

Second, the finding that confidence is best described by the relative strength of the evidence of the top two options might be related to other findings in multiple-alternative decision-making. In one experiment, the observer watched columns of bricks build up on the screen, and reported which column had the highest accumulation rate[42]. A heuristic model in which the observer makes a decision when the height of the tallest column exceeds the height of the next-tallest column by a fixed threshold captured the overall pattern of people's behavior. In a study on self-directed learning in a three-alternative categorization task, observers had to learn the category distributions by sampling from the feature space and receiving feedback. Instead of choosing the most informative samples, human observers chose ones for which the likelihood of two categories were similar, namely those located at boundaries between pairs of two categories[43]. This literature allows us to speculate that observers might decompose a multiple-alternative decision into several simpler (perhaps two-alternative) decisions. This notion is reminiscent of the concept in prospect theory that before a phase of evaluation, extremely unlikely outcomes might be first discarded in an "editing" phase[44]. Hence, an alternative interpretation of our results is that confidence reports deviate from the Bayesian confidence hypothesis (the Max model) because the observer estimates the probability of correct in a way that ignores the options that are discarded before final evaluation. In the Difference model, the least favorite option is not completely discarded because it decreases the posterior probabilities of the other two options (and thus their difference) by contributing to the normalization pool[45,46]. Therefore, we consider an extreme version of editing, the Ratio model, in which the least-favorite option does not even participate in normalization, and thus confidence solely depends on the likelihood ratio between the top two options. The Difference model and the Ratio model are not distinguishable in Experiment 1 and 2 (Supplementary Fig. 12). In Experiment 3, the Difference model has a slight advantage over the Ratio model by a group-summed AIC of 51 (95% CI [18, 90]). Testing variable numbers of categories within an experiment might help to differentiate between these two models.

We found that compared to the sensory noise, the noise associated with the computation of posterior probability plays a more important role in our task. This is consistent with the findings of a recent study[26]. The relative unimportance of sensory noise could be partly due to our experimental design, which used stimuli with strong signal strength (saturated color and unlimited duration). Differently from our study, Drugowitsch et al.[26] used an evidence accumulation task and further distinguished two types of decision noise: inference noise that was added with each new stimulus sample, and selection noise that was injected only once, right before the final response. Because our experiment only had one stimulus in each trial, it was not set up to distinguish these two sources of variability. While modeling decision noise using Dirichlet distribution was successful, we found that models in which the category mean is not known exactly but measured in a noisy fashion also fit the data in two experiments quite well (Diff-Sen-Mean model in Experiment 1 and Experiment 2; Supplementary Fig. 10). This is consistent with a recent finding that imperfect knowledge about the experimental parameters explained a significant portion of the behavioral variability in two-alternative decision tasks[47].

Across the three experiments, we did not find evidence that any of the heuristic models we tested outperformed the Difference model. In only one experiment, a heuristic model was indistinguishable from the Difference model (DistW-Sen-Mean model in Experiment 2; Supplementary Fig. 10B). These results are different from a recent study reporting that some probabilistic but heuristic models outperformed the Bayesian models in fitting confidence data in two-alternative tasks[25]. The causes of this discrepancy may lie in the experimental design. Adler and Ma manipulated different levels of uncertainty by presenting brief stimuli (50 ms) at various contrast levels. To perform the task optimally, observers had to track the sensory uncertainty varied

in a range (e.g., from 0.4 to 13% contrast in their main experiment) in a trial-by-trial fashion. Instead, we purposely reduced uncertainty by presenting all the stimuli at the highest achievable contrast with unlimited duration. In addition, the distributions of the categories were not presented explicitly in Adler and Ma's study whereas the distributions of the categories were presented throughout each trial in the present study. These factors may contribute to the fact that heuristic models performed better in Adler and Ma's study but not in our experiments.

The ΔAIC (relative to the Difference model) was smaller in Experiment 3 than in Experiment 1 and 2. Intuitively, this is not surprising, since when directly sampling from the stimulus distributions in Experiment 3, there were more target dots positioned at the far left and far right, compared to the target dots in Experiment 1 and 2. These trials would not have been informative to distinguish the models. All three models have the same prediction (high confidence) for far-left and the far-right locations, and this may lead to a smaller ΔAIC. To examine whether stimulus selection alone could account for the smaller AIC differences, we performed a model recovery analysis. We synthesized data based on the observers' best-fitted parameters of the Max, Difference and Entropy models, and we fitted the synthesized data with these three models. We found that the performance of the Max and the Difference models are closer in Experiment 3 than in Experiment 1 and 2, similar to the real data (Supplementary Fig. 3). Thus, stimulus selection alone can account for the smaller AIC differences.

Whereas we propose a theoretical framework for how decisions and confidence reports are computed in multi-alternative tasks, we are agnostic about how the decision-making process unfolds over time. Other models exist that consider the temporal dynamics of decision-making. In particular, drift-diffusion models and race models jointly account for accuracy and reaction times in many tasks[48]. Some studies have employed such accumulation models to account for confidence judgments[34,49–52]. However, these studies only considered confidence judgments in two-alternative decision tasks. Conceptually, our findings might be related to the "balance of evidence" (BoE) in Vickers and colleagues' work[51,53]. In a race model with two accumulators, they suggested that confidence is computed as the difference between the accumulated evidence of the two accumulators[51]. Vickers and Lee suggested that in theory, this idea could be extended to three-alternative tasks, but they speculated that confidence in the chosen category (option A) might be computed as the average of the confidence in comparing option A to option B and the confidence in comparing option A to option C[53]. This algorithm is more similar to the Max model than to the Difference model here: Assuming that A, B and C represent the evidence accumulated for each of the three categories, and A is the chosen category, confidence is computed as $c* = ((A−B) + (A−C))/2 = (3A−1)/2$. Then, confidence only depends on the chosen category A. It remains to be seen whether evidence accumulation models designed to explain decisions in multiple-alternative tasks (review in refs. [57,58]) could be extended to generate confidence reports that are consistent with our data and with the Difference model.

Do our results generalize beyond perceptual decision-making? In a two-alternative value-based decision task, observers reported confidence in a way that was similar to that in perceptual decision tasks:[10] When observers were asked to choose the good with the higher value, confidence increased with the posterior probability that a decision is correct, which in turn increased with the difference in value between the two goods. In addition, choice accuracy was higher in high-confidence trials then in low-confidence trials, reflecting observers' ability to evaluate their own performance. A recent study also reported that observers are able

to reflect on their decisions and report confidence in three-alternative value-based decision tasks[54]. Given that the computation of subjective value may involve a Bayesian inference process similar to that in perception[12], it might be worth investigating whether confidence reports in multiple-alternative value-based decisions also deviate from the Bayesian confidence hypothesis. The Difference model would predict that, confidence is determined by the difference between the probability that the chosen item is the most valuable and the probability that the next-best item is the most valuable.

How does the present study advance our understanding of the neural basis of confidence? Most neurophysiological studies of confidence have considered the neural activity that correlates with the probability of being correct as the neural representation of confidence (but see ref. [55]). Neural responses in parietal cortex[19], orbitofrontal cortex[18] and pulvinar[20] have been associated with that representation of confidence. These studies all used two-alternative decision tasks. Multiple-alternative decision tasks have been used in neurophysiological studies on non-human primates but not with the objective of studying confidence[46,56–58]. By utilizing multiple-alternative tasks, neural studies could dissociate the neural correlates of probability correct from that of the "difference" confidence variable in the Difference model, which according to our results, might be the basis of human subjective confidence. A potentially important difference between human and non-human animal studies is that in the latter, confidence is not explicitly reported but operationalized through some aspect of behavior, such as the probability of choosing a "safe" (opt-out) option[19,20,55,59,60], or the time spent on waiting for reward[18]. Thus, one should be careful when directly comparing these implicit reports with explicit confidence reports in human studies.

## Methods

**Setup.** Participants sat in a dimly lit room with the chin rest positioned 45 cm from the monitor. The stimuli and the experiment were controlled by customized programs written in Javascript. The monitor had a resolution of 3840 by 2160 pixels and a refresh rate of 30 Hz. The spectrum and the luminance of the monitor were measured with a spectroradiometer.

**Participants.** Thirteen participants took part in Experiment 1. Eleven participants took part in Experiment 2. Eleven participants took part in Experiment 3. All participants had normal or corrected-to-normal vision. The experiments were conducted with the written consent of each participant. The University Committee on Activities involving Human Subjects at New York University approved the experimental protocols.

**Stimulus.** On each trial, three categories of exemplar dots (375 dots per category) were presented along with one target dot, a black dot (Fig. 1a). The exemplar dots within a category were distributed as an uncorrelated, circularly symmetric Gaussian distribution with a standard deviation of 2° (degree visual angle) along both horizontal and vertical directions. Exemplar dots from the different categories were coded with different colors. The three colors were randomly chosen on each trial, and were equally spaced in Commission Internationale de l'Eclairage (CIE) $L*a*b*$ color space. The three colors were at a fixed lightness of $L* = 70$ and were equidistant from the gray point ($a* = 0$, and $b* = 0$).

In Experiment 1 and 3, the centers of the three categories were aligned vertically to the center of the screen, and were located at different horizontal positions (Fig. 1b). In four configurations, the horizontal positions of the centers of the three categories were (−3°, 0°, 3°), (−4°, 0°, 4°), (−3°, −2°, 3°), and (−3°, 2°, 3°), from the center of the screen respectively. In Experiment 2, the centers of the three categories varied on a 2-dimensional space (Fig. 1c). In four configurations, the horizontal positions of the centers of the three categories were (−2°, 0°, 2°), (−1.59°, 0°, 1.59°), (−2°, −2°, 2°), and (−2°, 2°, 2°), from the center of the screen, respectively. The vertical positions of the centers were (1.16°, −2.31°, 1.16°), (0.94°, −1.84°, 0.94°), (1.16°, 0°, 1.16°), and (1.16°, 0°, 1.16°) from the center of the screen respectively.

**Procedures.** We told participants that the three groups of exemplar dots represented a bird's eye view of three groups of people. The three groups contained equal numbers of people. The black dot (the target) is a person from one of the three groups, but we do not know the color of her/his T-shirt. We asked participants to categorize the target to one of the three groups based on the (position)

information conveyed by the dots, and report their confidence on a four-point Likert scale.

Each trial started with the onset of the stimulus and three rectangular buttons positioned at the bottom of the screen (Fig. 1a). On each trial, participants first categorized the target to one of the three groups (based on the position information conveyed by the dots) by using the mouse to click on one of the three buttons. After participants reported their decision, the three buttons were replaced by four buttons (labeled as "very unconfident", "somewhat unconfident", "somewhat confident", and "very confident") for participants to report their confidence on the decision they made. The stimuli were presented throughout each trial. Reaction time (for both category decision and confidence reports) was unlimited. After participants reported their confidence, all the exemplar dots and the rectangular buttons disappeared from the screen, and the next trial started after a 600 ms inter-trial-interval.

In Experiment 1, the vertical position of the target dot was sampled from a normal distribution (2° std), and the horizontal position of the target dot was sampled uniformly between the center of the leftmost and rightmost categories plus a 0.2° extension to the left and the right. In Experiment 2, the target dot was uniformly sampled from a circular area (2.6° radius) positioned at the center of the screen. No feedback was provided in Experiment 1 and Experiment 2.

In Experiment 3, in each trial, we randomly chose one of the three categories with equal probability as the true category. We then positioned the target dot by sampling from the distribution of the true category. A feedback regarding the true category was provided at the end of each trial: After participants reported their confidence, all exemplar dots disappeared except that the exemplar dots from the true category remained on the screen for an extra 500 ms. In each experiment, participants completed one 1-hr session (84 trials per configuration in Experiment 1 and 120 trials per configuration in Experiment 2 and 3). All the trials in one session were separated into eight blocks with equal number of trials. Different configurations were randomized and interleaved within each block.

Participants were well informed about the structure of the stimuli. We told observers that the distributions of the three groups are circular and symmetric, and the three groups have the same spread (standard deviation) throughout the experiments, and only differed in their centers. In Experiment 1 and 3, participants were informed that the centers of the three groups only varied horizontally.

**Models**. Generative model. The target belongs to category $C \in \{1, 2, 3\}$. The two-dimensional position $\mathbf{s}$ of a target in category $C$ is drawn from a two-dimensional Gaussian $p(\mathbf{s} \mid C) = N(\mathbf{s}; \mathbf{m}_C, \sigma_s^2 \mathbf{I})$, where $\mathbf{m}_C$ is the center of category $C$, $\sigma_s^2$ is the variance of the stimulus distribution, and $\mathbf{I}$ is the two-dimensional identity matrix. We assume that the observer makes a noisy sensory measurement $\mathbf{x}$ of the target position. We model the sensory noisy using a Gaussian distribution centered at $\mathbf{s}$ with covariance matrix $\sigma^2 \mathbf{I}$. Thus, the distribution of $\mathbf{x}$ given category $C$ is $p(\mathbf{x} \mid C) = N(\mathbf{x}; \mathbf{m}_C, (\sigma_s^2 + \sigma^2)\mathbf{I})$.

Inference on a given trial. We assume that the observer knows the mean and standard deviation of each category based on the exemplar dots, and that the observer assumes that the three categories have equal probabilities. The posterior probability of category $C$ given the measurement $\mathbf{x}$ is then $p(C \mid \mathbf{x}) \propto p(\mathbf{x} \mid C) = N(\mathbf{x}; \mathbf{m}_C, (\sigma_s^2 + \sigma^2)\mathbf{I})$. Instead of the true posterior $p(C \mid \mathbf{x})$, the observer makes the decisions based on $q(C \mid \mathbf{x})$, a noisy version of the posterior probability. We obtain a noisy posterior $q(C \mid \mathbf{x})$ by drawing from a Dirichlet distribution. The Dirichlet distribution is a generalization of the beta distribution. Just like the beta distribution is a continuous distribution over the probability parameter of a Bernoulli random variable, the Dirichlet distribution is a distribution over a vector that represents the probabilities of any number of categories. The Dirichlet distribution is parameterized as

$$p(\mathbf{q}|\mathbf{p}; \alpha) = \frac{1}{B(\alpha \mathbf{p})} \prod_{i=1}^{3} q_i^{\alpha p_i - 1}$$

$$B(\alpha \mathbf{p}) = \frac{\prod_{i=1}^{3} \Gamma(\alpha p_i)}{\Gamma\left(\alpha \sum_{i=1}^{3} p_i\right)}$$

$\Gamma$ represents the gamma function. $\mathbf{p}$ is a vector consisting of the three posterior probabilities, $\mathbf{p} = (p_1, p_2, p_3) = (p(C = 1 \mid \mathbf{x}), p(C = 2 \mid \mathbf{x}), p(C = 3 \mid \mathbf{x}))$. $\mathbf{q}$ is a vector consisting of the three posterior probabilities perturbed by decision noise, $\mathbf{q} = (q_1, q_2, q_3) = (q(C = 1 \mid \mathbf{x}), q(C = 2 \mid \mathbf{x}), q(C = 3 \mid \mathbf{x}))$. The expected value of $\mathbf{q}$ is $\mathbf{p}$. The concentration parameter $\alpha$ is a scalar whose inverse determines the magnitude of the decision noise; as $\alpha$ increases, the variance of $\mathbf{q}$ decreases. To make a category decision, the observer chooses the category that maximizes the posterior probability: $\hat{C} = \underset{C}{\operatorname{argmax}}\, q(C \mid \mathbf{x})$.

We considered three models of confidence reports. We first specify in each model an internal continuous confidence variable $c^*$. In the Max (maximum a posteriori) model, $c^*$ is the posterior probability of the chosen category: $c^* = q(C = \hat{C} \mid \mathbf{x})$. In the Difference model, $c^*$ is a difference: $c^* = q(C = \hat{C} \mid \mathbf{x}) - q(C = \hat{C}_2 \mid \mathbf{x})$, where $\hat{C}_2$ is the category with the second-highest

posterior probability. In the Entropy model, $c^*$ is the negative entropy of the posterior distribution: $c^* = \sum_{C=1}^{3} q(C \mid \mathbf{x}) \log q(C \mid \mathbf{x})$.

In each model, the internal confidence variable $c^*$ is converted to a four-point confidence report $c$ by imposing three confidence criteria $b_1$, $b_2$ and $b_3$. For example, $c = 3$ when $b_2 < c^* < b_3$. This implementation accommodated any type of mapping between the internal confidence variables $c^*$ and the four-level button press, as long as the reported levels monotonically increased with the internal confidence variables $c^*$. We also included a lapse rate $\lambda$ in each model; on a lapse trial, the observer presses a random button for both the decision and the confidence report. In addition to the models that included both sensory and Dirichlet decision noise, we took a factorial approach and tested various combinations of confidence model and sources of variability[61–63]. For each of the three main confidence models (Max, Difference and Entropy), we tested two reduced models by removing either the sensory noise (by setting $\sigma = 0$) or the decision noise (by setting $q(C \mid \mathbf{x}) = p(C \mid \mathbf{x})$) from the model, leading to nine models reported in Supplementary Figs. 4 and 5. In addition, we fitted these nine models with confidence reports only, without jointly fitting the category decisions (Supplementary Figs. 6 and 7). We also fitted the category decisions alone by three different models. These three models all chose the category with the highest posterior probability, but considered different sources of variability (sensory noise only, decision noise only, or both; Supplementary Fig. 8).

In addition to the nine models reported in Supplementary Fig. 4 and Supplementary Fig. 5, we furthermore tested 21 alternative models (Supplementary Fig. 10), including Bayesian models with various sources of variability and heuristic models that made decisions without computing posterior probability. The details of these models are described in Supplementary Information.

Response probabilities. So far, we have described the mapping from a measurement $\mathbf{x}$ to a decision $\hat{C}$ and a confidence report $c$. The measurement, however, is internal to the observer and unknown to the experimenter. Therefore, to obtain model predictions for a given parameter combination ($\sigma, \alpha, b_1, b_2, b_3, \lambda$), we perform a Monte Carlo simulation. For every true target position $\mathbf{s}$ that occurs in the experiment, we simulated a large number (10,000) of measurements $\mathbf{x}$. For each of these measurements, we compute the posterior $p(C \mid \mathbf{x})$, add decision noise to obtain $q(C \mid \mathbf{x})$, and finally obtain a category decision $\hat{C}$ and a confidence report $c$. Across all simulated measurements, we obtain a joint distribution $p(\hat{C}, c | \mathbf{s}; \sigma, \alpha, b_1, b_2, b_3, \lambda)$ that represents the response probabilities of the observer.

Model fitting and model comparison. We denote the parameters ($\sigma, \alpha, b_1, b_2, b_3, \lambda$) collectively by $\theta$. We fit each model to individual-subject data by maximizing the log likelihood of $\theta$, $\log L(\theta) = \log p(\text{data}|\theta)$. We assume that the trials are conditionally independent. We denote the target position, category response, and four-point confidence report on the ith trial by $\mathbf{s}_i$, $\hat{C}_i$, and $c_i$, respectively. Then, the log likelihood becomes

$$\log L(\theta) = \log \prod_i p(\hat{C}_i, c_i | \mathbf{s}_i, \theta) = \sum_i \log p(\hat{C}_i, c_i | \mathbf{s}_i, \theta),$$

where $p(\hat{C}_i, c_i | \mathbf{s}_i, \theta)$ is obtained from the Monte Carlo simulation described above. We optimized the parameters for each individual using a new method called Bayesian Adaptive Direct Search[64]. We used AIC for model comparison. To report the AIC, we computed the AIC for each individual and then summed the AIC across participants. The confidence interval of the group-summed AIC was estimated by bootstrapping. We also reported BIC in Supplementary Information.

**Parameterization**. The three main models (Max, Difference and Entropy models reported in Fig. 4) have the same set of free parameters including the magnitude of sensory noise ($\sigma$), the magnitude (concentration parameter) of decision noise ($\alpha$), three boundaries for converting internal continuous confidence variable to button press ($b_1$, $b_2$, $b_3$) and a lapse rate $\lambda$. For each of the three models, we tested two versions of the reduced models (Supplementary Figs. 4–6). In one version, we kept the sensory noise ($\sigma$) in the model while removing the decision noise ($\alpha$). In the other version we kept the decision noise ($\alpha$) in the model while removing the sensory noise ($\sigma$). The details of other alternative models are described in Supplementary Information.

**Model recovery**. To evaluate our ability to distinguish the three models, we performed a model recovery analysis. Based on the design of each experiment (including the stimulus distributions, target locations and the number trials), we synthesized a dataset based on the best-fit parameters of each participant. We then fit each of the datasets with the three models. Supplementary Fig. 3 illustrates the results summed over all participants in each experiment.

**Data visualization**. For Experiments 1 and 3, we used a sliding window to visualize the psychometric curves, defined as the confidence ratings as a function of horizontal location of the target dot. The sliding window had a width of 0.6°. We moved the window horizontally (in a step of 0.1°) from the left to the right of the screen center. At each step, we computed mean confidence rating by averaging the confidence reports $c$ of all the trials that fell within the window (based on the horizontal target location of each trial). We first applied this procedure to individual data, and then averaged the individual psychometric curves across subjects

(Fig. 3b, Supplementary Figs. 2, 7, 9 and 11). For Experiment 1, we visualized the data ranging from −3.5° to +3.5° from the screen center. For Experiment 3, we visualized the data ranging from −5° to +5° from the center. These ranges were chosen so that each steps along the curves in Fig. 3b, Supplementary Figs. 2, 7, 9 and 11 contained at least five trials per subject on average. To visualize the model fit, we sampled a series of evenly spaced target dot locations along the horizontal axis (in a step of 0.1°), and we used the best-fitting parameters to compute the confidence reports predicted by the models for each target location. We then used the same procedure (a sliding window) to compute the mean confidence rating predicted by the models (the model-fit curves in Fig. 3b, Supplementary Figs. 2, 7, 9 and 11).

For Experiment 2, the "psychometric curve" became a heat map in a two-dimensional space (Fig. 5). We tiled the two-dimensional space with non-overlapped hexagonal spatial windows (with a radius of 0.25°) positioned from −3° to +3° (Fig. 5a) along both horizontal and vertical axis. To compute the mean confidence rating for each hexagonal window, we averaged the confidence ratings across all the trials fell within that window for each participant. If the number of trials was zero among all the participants for a window, that window was left as white in Fig. 5a. To visualize the model fit, we used the best-fitting parameters and computed the confidence reports predicted by the models for an array of target locations (a grid tiling the two-dimensional space with a step of 0.1° along both horizontal and vertical axis). The predicted confidence reports were then averaged within each hexagonal window.

**Reporting summary**. Further information on research design is available in the Nature Research Reporting Summary linked to this article.

## Data availability
The data that support the findings of this paper are available on https://github.com/hsinhungli/confidence-multiple-alternatives

## Code availability
The analysis code used in this paper is available on https://github.com/hsinhungli/confidence-multiple-alternatives

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

## Acknowledgements

We thank members of the Ma Lab, Hui-Kuan Chung, Rachel Denison, and Michael Landy for helpful comments on the manuscript.

## Author contributions

H.-H. L. and W.J.M. designed the experiment. H.-H. L. performed the experiment and analyzed the data. H.-H. L. and W.J.M. wrote the manuscript.

## Competing interests

The authors declare no competing interests.
