## [Peer Review File · Nature Communications]

Reviewers' Comments:

Reviewer #1:

Remarks to the Author:

This paper reports a series of behavioral experiments using 3-alternative choices to demonstrate that human confidence judgments are best modeled as reporting the difference between the highest and second-highest probability. This model outperformed the standard "Bayesian confidence hypothesis" (that confidence judgments report the posterior probability of being correct) and an entropy-based hypothesis.

I found the results very compelling. The model-fitting and comparison techniques are rigorous, the experimental analysis is thorough, and the paper is clearly written.

A few technical points:

Eq. 1 should have different notation for C (e.g., C') in the numerator and denominator. Also, s should be marginalized out in this expression; the correct expression is $P(C|x) = P(x|C)P(C)/P(x)$.

The notation used in the Dirichlet distribution on line 436 needs to be modified slightly, because on the left hand side there's P_noisy but on the right hand side there's P_n. Also, you need to state that gamma represents the gamma function.

It is stated that alpha inverse determines the magnitude of the decision noise. However, this is a little confusing, because entropy actually goes to 0 as alpha approaches 0. It's important to distinguish here between bias and variance. As alpha approaches 0, the samples will have high bias but low variance.

Possibly relevant:

I was curious whether one could capture these patterns using a model in which a small number of samples are drawn from the posterior and the Bayesian confidence hypothesis is implemented on this Monte Carlo estimate of the posterior. This is relevant because a number of authors have suggested that people use such stochastic approximations. I implemented a simple simulation of this model, and found that in fact $P(\max)$ *decreases* with the difference between the highest and second-highest probabilities. Thus, at least in my simulations, this "stochastic Bayesian confidence hypothesis" cannot explain the data.

Minor comments:

line 437: "p is a vector consists" -> "p is a vector consisting of"

line 501: "of all the trials fell" -> "of all the trials that fell"

Reviewer #2:

Remarks to the Author:

In this paper Li and Ma extend the investigation of decision confidence beyond the classical 2-choice case to a 3-choice scenario. This seemingly minor adjustment proves critical for separating distinct models of how confidence relates to the computation of posterior probabilities (a key property of the leading "Bayesian confidence hypothesis"). The authors devise an innovative task to cleanly measure

subjects' choices and confidence in decisions about the category membership of noisy samples of evidence, and present a rigorous model comparison that indicates subjects' confidence ratings do not follow the BCH. Instead, the patterns of confidence ratings are better accommodated by a model that computes the difference in evidence between the best and next best options. This is an intriguing result, and as the authors point out in the discussion, suggests that the function of confidence may be to optimise a best-possible choice, rather than accurately reflect the evidence for the best option.

I think this will be an important and influential paper for the field and the results are clear and compelling. I have relatively minor comments.

1) The "Difference" model is highly reminiscent of the Vickers balance of evidence model, in which confidence is based on the difference between the winning and next-best losing accumulator in a race model (Vickers, 1979, *Decision Processes in Visual Perception*). This was once a dominant model of confidence in perceptual decision-making prior to the surge of interest in the BCH so I find it odd that it is not cited or discussed.

2) I found the introduction and setup of Experiment 1 somewhat confusing. On line 46 it is said while the field favours the BCH there is "no evidence to rule out the possibility that confidence is affected by unchosen options", and on line 113 "confidence is never influence by the posterior probabilities of the categories that were not chosen". Maybe it's just me, but I found this hard to get my head around even though I am fairly up to speed with the BCH. I think the issue is that of course Bayesian confidence is (implicitly) influenced by the unchosen options simply because these categories exist and they contribute to the normalisation step. In other words, if category number is variable, then just a single exemplar from a new category (or instructions that a new category is in play) should be sufficient to decrease confidence even in the Max observer. Therefore strong statements about the absence of influence of unchosen options are ambiguous – the authors mean the influence of unchosen evidence on normalised confidence, but this is subtle and could be misleading. I suggest finding a way to more gently introduce the reader to this critical distinction (perhaps with an intuitive example of the effects of normalisation versus the effects of unchosen evidence). In addition, at this point in the paper I found myself imagining a non-Bayesian version of the Max model in which confidence was left unnormalized and not even sensitive to the presence of other categories. It wasn't until I got to the Discussion that I learnt this type of model had also been considered (the Ratio model). Perhaps it could be flagged earlier that models of this sort will be considered in the Discussion?

3) Line 332, "it is unknown how observers compute confidence when there are more than two goods". This was looked at in Folke et al. (2017) *Nature Human Behaviour* – though I don't think the three-choice aspect of the task was directly linked to confidence in their paper. The data are available online though, which may allow the authors to test the predictions of the Difference model in value-based decisions.

4) Were subjects instructed about the spatial clustering of people, or was this left implicit?

Reviewer #3:

Remarks to the Author:

In this study, Li and Ma study psychological processes underlying confidence reports when participants make decisions based on multiple alternatives. Based on a series of behavioral experiments and quantitative analyses the authors argue that confidence reports after making decisions violate the Bayesian confidence hypothesis. This was possible thanks to the fact that the decision task was based

on more than two alternatives, which allowed the authors to disentangle differentiate between different possible strategies that human participants may use to report confidence after a forced choice. In my opinion, the authors tackle an important and timely question, which might be of interest to a wide audience in psychology and decision sciences in general. There are however important concerns that the authors may want to clarify before I am convinced of the strong claims (starting from the title) that the Bayesian confidence hypothesis is indeed violated.

1) If I understood the description of the task correctly, the participants are first asked to make a categorical choice, and then are asked to rate their confidence based on their decision i.e. two "choices" in sequence. However, the authors use the quantitative strategy of jointly fitting choices and confidence, where it is assumed that categorical choices are Bayesian, but the different confidence reports are different depending on the assumed strategy. I am not convinced that jointly fitting choices and confidence is entirely correct as these are not joint responses as it is the case of, for instance, choice-RT distributions, or as Ma and colleagues in fact used it in previous studies where confidences and choices are part of the same response. This is problematic in light of the results shown in Supplementary Figure 2, where jointly fitting choices and confidence appear to harm the categorical choice fits (more evidently in the entropy model). Here, I would like to see a couple of things: First, how the categorical choice fits look in the absence of joint confidence fits. In this case which of the three noise models fits the data best quantitatively?. Second and more importantly, model comparisons when confidence ratings are fit "independently" (please report qualitative fits as well as AICs and BICs for these analyses). With the current experimental setup, there is no reason why confidence should harm categorical choice fits. If the authors would insist on this approach, then I would strongly propose that the authors follow an experimental paradigm where choices and confidence are indeed jointly reported and then repeat the model comparison approach as currently proposed by the authors.

2) Before accepting the rather strong claim that the Bayesian confidence hypothesis is in fact violated, the two following points must be clarified: First, I would like to see the BIC information as reported for the AIC in Supplementary Table 1 and Supplementary Figure 7. Based on the results observed in Supplementary Figure 5, it appears as if there is no convincing evidence that Difference model fits the data better than the Max model, especially for experiment 3, which is the only experiment wherein fact the participants are allowed to learn about the probability of being correct. Second, the quantitative comparisons are based on the strong assumption that categorization is Bayesian, but then test two non-Bayesian alternative models when reporting confidence. Interestingly, Ma and colleagues have shown in the recent past that these types of categorization models do not follow the optimal Bayesian rules. Is there any non-Bayesian model that fits better the model the categorization data than the Bayesian model? If this is the case, how are the conclusions of the authors affected. Related to this, it has been shown in the past that humans might be non-optimal Bayesian agents (e.g., see studies by Daniel Benjamin), where it might be still the case that participants try to use Bayesian rules, but weighting of likelihoods and priors is not optimal, and it could be that at the confidence stage they use for instance the Max model. In summary, the agents could still be Bayesian but suboptimal, and this could change the conclusions put forward by the authors in this study.

3) In the difference model, confidence is based on the difference between the chosen and second best alternative. I would be curious to a model that compares the best with the average of the non-chosen alternatives. Maybe I am missing something this is redundant, but I find it interesting that categorization is based on the effortful optimal evaluation all the alternatives during categorization, and then possible valuable information is disregarded during the confidence ratings (I accept that this is not entirely true as the posterior over categories is still affected by the worst alternative).

4) It is not entirely clear to me what how the current findings generalize to estimation tasks. If I

understand correctly, during estimation tasks participants are assumed to generate an estimate based on a continuous posterior distribution over a continuous estimation scale and then generate a rating based on this posterior distribution. In this case, what would be second best alternative, more specifically for the case when the rating scale is also continuous?

5) In line 94, I think the first p should not be bolded.

6) In Figure 5, please make more salient the center of the three categories.

Reviewer #1 (Remarks to the Author):

This paper reports a series of behavioral experiments using 3-alternative choices to demonstrate that human confidence judgments are best modeled as reporting the difference between the highest and second-highest probability. This model outperformed the standard "Bayesian confidence hypothesis" (that confidence judgments report the posterior probability of being correct) and an entropy-based hypothesis.

I found the results very compelling. The model-fitting and comparison techniques are rigorous, the experimental analysis is thorough, and the paper is clearly written.

A few technical points:

Eq. 1 should have different notation for C (e.g., C') in the numerator and denominator. Also, s should be marginalized out in this expression; the correct expression is $P(C|x) = P(x|C)P(C)/P(x)$.

We have changed the notation (line 86).

The notation used in the Dirichlet distribution on line 436 needs to be modified slightly, because on the left hand side there's P_noisy but on the right hand side there's P_n. Also, you need to state that gamma represents the gamma function.

To distinguish the true posterior and the noisy posterior, we now denote the noisy posterior by q throughout the manuscript.

We now also state that the gamma symbol represents the gamma function (line 539).

It is stated that alpha inverse determines the magnitude of the decision noise. However, this is a little confusing, because entropy actually goes to 0 as alpha approaches 0. It's important to distinguish here between bias and variance. As alpha approaches 0, the samples will have high bias but low variance.

This is not correct. We hypothesize that the noisy posterior $\mathbf{q}=(q_1, q_2, q_3)$ follows a Dirichlet distribution, $\mathbf{q} \sim \text{Dir}(\mathbf{p}, \alpha)$. (The parameterization of this distribution is described in lines 538-545.) Here, $\mathbf{p}=(p_1, p_2, p_3)$ is the true posterior and α is a scalar. The expected value of \mathbf{q} is equal to \mathbf{p} , regardless of α .

$$E[q_i] = p_i$$

Thus, there is no bias, regardless of α . Instead, α determines the magnitude of the decision noise. The larger the alpha the lower the variance of \mathbf{q} :

$$\text{var}(q_i) = \frac{p_i(1-p_i)}{\alpha+1}$$

We elaborate on these points in lines 542-545.

Possibly relevant:

I was curious whether one could capture these patterns using a model in which a small number of samples are drawn from the posterior and the Bayesian confidence hypothesis is implemented on this Monte Carlo estimate of the posterior. This is relevant because a number of authors have suggested that people use such stochastic approximations. I implemented a simple simulation of this model, and found that in fact $P(\text{max})$ *decreases* with the difference between the highest and second-highest probabilities. Thus, at least in my simulations, this "stochastic Bayesian confidence hypothesis" cannot explain the data.

This is an interesting proposal. The Dirichlet distribution that we used is the conjugate prior for the multinomial distribution. We suspected that modeling the noisy posterior as arising from Monte Carlo samples of the true posterior would lead to results that are similar to modeling the noisy posterior as being drawn from a Dirichlet distribution.

We re-implemented the Max, Difference and Entropy models, but we modeled the inference noise by Monte Carlo simulation (by drawing samples based on the true posterior), instead of using the Dirichlet distribution. The number of samples was a free parameter. The noisy posterior q is determined by the outcome of Monte Carlo simulation in these models (see details of the models in the section '*Sampling noise*' in **Supplementary Information**). We found that the Difference model outperformed the Max and the Entropy models, consistent with the results obtained using Dirichlet decision noise (lines 210-213; Diff-Samp, Max-Samp, Ent-Samp, Diff-Sen-Samp, Max-Sen-Samp and Ent-Sen-Samp models in **Supplementary Figure 10** and **Supplementary Table 1-2**). Thus, our main conclusions are not affected. In addition, the Diff-Sen-Dir model and the Diff-Sen-Samp model are indistinguishable in Experiment 1 and Experiment 3. In Experiment 2, the Diff-Sen-Dir model slightly outperforms the Diff-Sen-Samp model by AIC score of 8.3 ± 4.8 (mean \pm s.e.m.) Overall, these results confirm that modeling the noisy posterior as arising from Monte Carlo samples of the true posterior is similar to modeling the noisy posterior as being drawn from a Dirichlet distribution.

Minor comments:

line 437: "p is a vector consists" -> "p is a vector consisting of"

We corrected this (line 540).

line 501: "of all the trials fell" -> "of all the trials that fell"

We corrected this (line 618).

Reviewer #2 (Remarks to the Author):

In this paper Li and Ma extend the investigation of decision confidence beyond the classical 2-choice case to a 3-choice scenario. This seemingly minor adjustment proves

critical for separating distinct models of how confidence relates to the computation of posterior probabilities (a key property of the leading “Bayesian confidence hypothesis”). The authors devise an innovative task to cleanly measure subjects’ choices and confidence in decisions about the category membership of noisy samples of evidence, and present a rigorous model comparison that indicates subjects’ confidence ratings do not follow the BCH. Instead, the patterns of confidence ratings are better accommodated by a model that computes the difference in evidence between the best and next best options. This is an intriguing result, and as the authors point out in the discussion, suggests that the function of confidence may be to optimise a best-possible choice, rather than accurately reflect the evidence for the best option.

I think this will be an important and influential paper for the field and the results are clear and compelling. I have relatively minor comments.

1) The “Difference” model is highly reminiscent of the Vickers balance of evidence model, in which confidence is based on the difference between the winning and next-best losing accumulator in a race model (Vickers, 1979, *Decision Processes in Visual Perception*). This was once a dominant model of confidence in perceptual decision-making prior to the surge of interest in the BCH so I find it odd that it is not cited or discussed.

We thank the reviewer for pointing out this reference. We now discuss Vickers’ balance of evidence, and the accumulation models in general in the Discussion (lines 398-417).

2) I found the introduction and setup of Experiment 1 somewhat confusing. On line 46 it is said while the field favours the BCH there is “no evidence to rule out the possibility that confidence is affected by unchosen options”, and on line 113 “confidence is never influenced by the posterior probabilities of the categories that were not chosen”. Maybe it’s just me, but I found this hard to get my head around even though I am fairly up to speed with the BCH. I think the issue is that of course Bayesian confidence is (implicitly) influenced by the unchosen options simply because these categories exist and they contribute to the normalisation step. In other words, if category number is variable, then just a single exemplar from a new category (or instructions that a new category is in play) should be sufficient to decrease confidence even in the Max observer. Therefore strong statements about the absence of influence of unchosen options are ambiguous – the authors mean the influence of unchosen evidence on normalised confidence, but this

is subtle and could be misleading. I suggest finding a way to more gently introduce the reader to this critical distinction (perhaps with an intuitive example of the effects of normalisation versus the effects of unchosen evidence). In addition, at this point in the paper I found myself imagining a non-Bayesian version of the Max model in which confidence was left unnormalized and not even sensitive to the presence of other categories. It wasn't until I got to the Discussion that I learnt this type of model had also been considered (the Ratio model). Perhaps it could be flagged earlier that models of this sort will be considered in the Discussion?

We have changed our wording describing the influence of the unchosen options (lines 47-48; lines 111-113). We now explicitly state that the unchosen options contribute to the computation of the posterior probability distribution, and thus "implicitly" affect confidence (lines 129-134). We also clarify that in the readout stage, the BCH does not use the posterior probabilities of the unchosen options (lines 137-138).

We now preview (in lines 132-134) that the Ratio model will be considered in the Discussion.

3) Line 332, "it is unknown how observers compute confidence when there are more than two goods". This was looked at in Folke et al. (2017) Nature Human Behaviour – though I don't think the three-choice aspect of the task was directly linked to confidence in their paper. The data are available online though, which may allow the authors to test the predictions of the Difference model in value-based decisions.

We thank the reviewer for pointing out this reference that we missed. We now address Folke et al. (2017) in lines 424-431. To directly compare the computation of confidence in value-based decisions to the current findings, one has to formulate the value-based decision process in a Bayesian framework. We point this out as a future direction of this line of research (lines 424-431).

4) Were subjects instructed about the spatial clustering of people, or was this left implicit?

The participants were well informed regarding the structure of the stimuli: We told observers that the distributions of the three groups are circular and symmetric, and the three groups have the same spread (standard deviation) and only differed in their

centers. In Experiment 1 and 3, participants were informed that the centers of the three groups only varied horizontally. (lines 514-518).

Reviewer #3 (Remarks to the Author):

In this study, Li and Ma study psychological processes underlying confidence reports when participants make decisions based on multiple alternatives. Based on a series of behavioral experiments and quantitative analyses the authors argue that confidence reports after making decisions violate the Bayesian confidence hypothesis. This was possible thanks to the fact that the decision task was based on more than two alternatives, which allowed the authors to disentangle differentiate between different possible strategies that human participants may use to report confidence after a forced choice. In my opinion, the authors tackle an important and timely question, which might be of interest to a wide audience in psychology and decision sciences in general. There are however important concerns that the authors may want to clarify before I am convinced of the strong claims (starting from the title) that the Bayesian confidence hypothesis is indeed violated.

1) If I understood the description of the task correctly, the participants are first asked to make a categorical choice, and then are asked to rate their confidence based on their decision i.e. two “choices” in sequence. However, the authors use the quantitative strategy of jointly fitting choices and confidence, where it is assumed that categorical choices are Bayesian, but the different confidence reports are different depending on the assumed strategy. I am not convinced that jointly fitting choices and confidence is entirely correct as these are not joint responses as it is the case of, for instance, choice-RT distributions, or as Ma and colleagues in fact used it in previous studies where confidences and choices are part of the same response. This is problematic in light of the results shown in Supplementary Figure 2, where jointly fitting choices and confidence appear to harm the categorical choice fits (more evidently in the entropy model). Here, I would like to see a couple of things: First, how the categorical choice fits look in the absence of joint confidence fits. In this case which of the three noise models fits the data best quantitatively?. Second and more importantly, model comparisons when confidence ratings are fit “independently” (please report qualitative fits as well as AICs and BICs for these analyses). With the current experimental setup, there is no reason why confidence should harm categorical choice fits. If the authors would insist on this approach, then I would strongly propose that the authors follow an experimental

paradigm where choices and confidence are indeed jointly reported and then repeat the model comparison approach as currently proposed by the authors.

The reviewer is correct that the reason that Entropy model has a poor fit for the categorization data is that the model was used to jointly fit the categorization and the confidence data, which the Entropy model performed very bad at (addressed in (b) below). The reviewer therefore suggested fitting the categorization data and the confidence data separately.

(a) Fitting the categorization data by themselves is not useful to distinguish the models, because all three models (Max, Difference and Entropy) used the same rules to categorize the target: “choose the category with the highest posterior” (lines 129-130; lines 564-568). We now show the fit of that single collapsed model to the categorization data alone in **Supplementary Figure 8** and **Supplementary Figure 9**. The model fit the data well. We also tested which type of noise (sensory vs. Dirichlet decision noise) can better fit the category report, without fitting the confidence reports. We found that the decision noise plays a more important role in explaining the data (lines 202-209; **Supplementary Figure 8** and **Supplementary Figure 9**), similar to what we found when fitting the confidence reports only, or jointly fitting category decisions and confidence.

(b) We also fit the confidence data by themselves, without fitting the categorization reports. We found that the Difference model outperformed the other two models, and the performance of the Entropy model is the worst. The decision noise had a greater impact on the model fit compared to the sensory noise. These results are the same as what we found when jointly fitting the category decision and confidence reports (**Supplementary Figure 6**).

2) Before accepting the rather strong claim that the Bayesian confidence hypothesis is in fact violated, the two following points must be clarified: First, I would like to see the BIC information as reported for the AIC in Supplementary Table 1 and Supplementary Figure 7.

We had BIC reported in **Supplementary Figure 5**. We now added a table reporting BIC scores, **Supplementary Figure 2**.

Based on the results observed in Supplementary Figure 5, it appears as if there is no convincing evidence that Difference model fits the data better than the Max model, especially for experiment 3, which is the only experiment wherein fact the participants are allowed to learn about the probability of being correct. Second, the quantitative comparisons are based on the strong assumption that categorization is Bayesian, but then test two non-Bayesian alternative models when reporting confidence. Interestingly, Ma and colleagues have shown in the recent past that these types of categorization models do not follow the optimal Bayesian rules. Is there any non-Bayesian model that fits better the model the categorization data than the Bayesian model? If this is the case, how are the conclusions of the authors affected.

(a) The reviewer is right that the delta AIC was smaller in Experiment 3 than those in Experiment 1 and 2. Even so, in Experiment 3, only 2 out of 11 subjects show negative delta AIC when comparing the Max to the Difference models (and only 1 out of 11 when the confidence data is fitted alone; **Supplementary table 1-3**). We don't think that this is due to the influence of the feedback. To examine this, we performed model recovery analysis for all three experiments using the experimental procedures and number of trials matching our experiments. Model recovery analysis showed that the delta AIC would be smaller in Experiment 3 than those in Experiment 1 and 2 (see **Model Recovery in Methods; Supplementary Figure 3**), mimicking our data. Thus, we think that the smaller delta AIC in Experiment 3 may be due to the selection of the target locations. When directly sampling from the stimulus distribution in Experiment 3, many target dots were positioned at the far-left or far-right locations, where all the confidence models have the same predictions (high confidence). We now discuss this in lines 385-397.

(b) Indeed, in previous work, our lab has reported that in some visual categorization tasks, observers' performance is better fitted by heuristic models than by Bayesian models. In the revision, we have tested nine heuristic models. Similar to Adler and Ma, these heuristic models made decisions and confidence reports based on the noisy measurement of the target and the stimuli, without computing posterior probabilities. These models are:

Distance model with sensory noise (Dist-Sen)

Distance model with noisy measurement of category mean (Dist-Mean)

Distance mode with sensory noise and noisy measurement of category mean (Dist-Sen-Mean)

Weighted distance model with sensory noise (DistW-Sen)

Weighted distance model with noisy measurement of category mean (DistW-Mean)

Weighted distance model with sensory noise and noisy measurement of category mean (DistW-Sen-Mean)

Distance-to-bound models with sensory noise (Bound-Sen)

Distance-to-bound models with noisy measurement of category mean (Bound-Mean)

Distance-to-bound models with sensory noise and noisy measurement of category mean (Bound-Sen-Mean)

In short, in the Distance model, the observer computes the distances between the noisy measurement of the target location and each of the three group centers. The observers categorize the target as belonging to the closest group center. Confidence is computed as the difference between the two shortest distances. The Weighted distance model extends the Distance model by allowing confidence to be computed as a linear combination of the three distances. The Distance-to-Bound model is inspired by Kepecs et al., 2008. Observers report confidence based on the distance between the target dot and the decision boundary (the midpoint) between the top-two categories. The details of these heuristic models are described in **Supplementary Information**.

We found that the heuristic models performed worse than the Difference model. These results are reported in lines 215-219, **Supplementary Table 1-2**, and **Supplementary Figure 10**. We discuss possible reasons that our results are different from the previous study (Adler and Ma, 2018) in lines 370-384.

Related to this, it has been shown in the past that humans might be non-optimal Bayesian agents (e.g., see studies by Daniel Benjamin), where it might be still the case that participants try to use Bayesian rules, but weighting of likelihoods and priors is not optimal, and it could be that at the confidence stage they use for instance the Max model. In summary, the agents could still be Bayesian but suboptimal, and this could change the conclusions put forward by the authors in this study.

Although the analysis above suggests that models employing the Bayesian rule outperformed other (heuristic) models, it does not mean that they are perfect. In particular, we found strong evidence for variability in the decision (inference) stage [e.g.,

Supplementary Figure 4-9], consistent with recent findings that investigated the sources of suboptimality in perceptual decision-making (Drugowitsch et al., 2016; Shen and Ma, 2019; Stengard and van den Berg, 2019). Thus, we agree with the reviewer that observers' behavior seems to be Bayesian but suboptimal. (We are not able to find the studies by Daniel Benjamin. We are happy to include the references if the reviewer provides the reference).

To further investigate the source of behavioral variability (or suboptimality) for a Bayesian observer, we now consider two additional sources of variability (or suboptimality):

- (a) Observers perform Bayesian inference but with imperfect (noisy) measurements of the centers of the three categories.
- (b) The noisy posterior is a sample from the true posterior (also in response to reviewer 1's comment).

We paired these two forms of variability with the previous confidence models, and we considered sensory noise being absent or present, leading to 12 additional models. The details of these models are described in **Supplementary Information**. We found that the Difference model still performs better than the other models (**Supplementary Figure 10**). We discuss these results in the paragraph in lines 210-213 and lines 363-369.

3) In the difference model, confidence is based on the difference between the chosen and second best alternative. I would be curious to a model that compares the best with the average of the non-chosen alternatives. Maybe I am missing something this is redundant, but I find it interesting that categorization is based on the effortful optimal evaluation all the alternatives during categorization, and then possible valuable information is disregarded during the confidence ratings (I accept that this is not entirely true as the posterior over categories is still affected by the worst alternative).

A model that compares the best with the average of the two non-chosen alternatives will be equivalent to the Max model. Because the three posteriors sum up to 1, the sum (and the average) of the two lower posteriors is determined by the highest posterior. Thus, such model is equivalent to the Max model. Specifically, when $\mathbf{p} = (p_1, p_2, p_3) = (p(C = 1 | \mathbf{x}), p(C = 2 | \mathbf{x}), p(C = 3 | \mathbf{x}))$ and $p_1 \geq p_2 \geq p_3$, the internal confidence variable is computed as:

$$\begin{aligned}
c^* &= p_1 - (p_2 + p_3) / 2 \\
&= p_1 - (1 - p_1) / 2 \\
&= \frac{3}{2} p_1 - \frac{1}{2}
\end{aligned}$$

The prediction of this model solely depends on p_1 , same as the Max model. We added a note about this in lines 121-122. We now also emphasize that the unchosen categories contribute to the decisions and confidence reports 'implicitly' by contributing to the normalization (lines 129-134).

4) It is not entirely clear to me what how the current findings generalize to estimation tasks. If I understand correctly, during estimation tasks participants are assumed to generate an estimate based on a continuous posterior distribution over a continuous estimation scale and then generate a rating based on this posterior distribution. In this case, what would be second best alternative, more specifically for the case when the rating scale is also continuous?

(a) We had addressed this point in the paragraph in lines 285-300. In a continuous estimation task, if the posterior distribution is unimodal (e.g., Gaussian), the best option is the peak of the distribution, and the second-best option is a point right next to the peak. The difference between these two points would approximate the curvature of the posterior distribution at its peak. This 'difference' is monotonically related to the inverse variance of the posterior distribution. Thus, the Difference model is consistent with an earlier model in which confidence is based on the precision parameter of the posterior distribution in estimation tasks (van den Berg, Yoo and Ma, 2017). We think that it is an open question regarding how confidence is reported when posterior is not unimodal in a continuous estimation task. For example, if the posterior distribution is bimodal, with the modes slightly different in height, the variance of the posterior is dominated by the separation between the modes, whereas the Difference model will use the difference in height for confidence reports (see lines 285-300).

(b) We fit the model by using three boundaries on the confidence variables derived from the posterior distributions. As the three boundaries were free parameters, this implementation accommodated any type of mapping between the confidence variables and the four-level button press, as long as the reported levels monotonically increased with the confidence variables. We believe that the results won't change under continuous rating scale as long as the rating monotonically increased with observers'

internal confidence representation. This point is addressed at (lines 105-107 and lines 554-556).

5) In line 94, I think the first p should not be bolded.

The p (now in line 93) should be bolded because it represents a vector.

6) In Figure 5, please make more salient the center of the three categories.

We replaced the figure by one in which we made the center of the three categories more salient.

Reviewers' Comments:

Reviewer #1:

Remarks to the Author:

The authors have done a good job addressing my comments (and pointing out an error in my reasoning). I'm happy to sign off on the paper.

Reviewer #2:

Remarks to the Author:

The authors have addressed all my comments, and I have no further concerns. I congratulate them on an impressive paper.

On line 425 "reflect" should be "reflect on".

Reviewer #3:

Remarks to the Author:

The authors have addressed all my comments rigorously. I commend the authors for this inspiring work and recommend this work for publication.

R.P.